# HC-HA/PTX3 from Human Amniotic Membrane Induced Differential Gene Expressions in DRG Neurons: Insights into the Modulation of Pain

**DOI:** 10.3390/cells13221887

**Published:** 2024-11-15

**Authors:** Shao-Qiu He, Chi Zhang, Xue-Wei Wang, Qian Huang, Jing Liu, Qing Lin, Hua He, Da-Zhi Yang, Scheffer C. Tseng, Yun Guan

**Affiliations:** 1Department of Anesthesiology and Critical Care Medicine, School of Medicine, Johns Hopkins University, Baltimore, MD 21205, USA; shaoqiuh@hotmail.com (S.-Q.H.); czhang147@usf.edu (C.Z.); xueweiwang@usf.edu (X.-W.W.); windysalad@hotmail.com (Q.H.); jliu307@jhu.edu (J.L.); qlin2@jhmi.edu (Q.L.); 2BioTissue, Inc., Miami, FL 33126, USA; hhe@biotissue.com (H.H.); stseng@biotissue.com (S.C.T.); 3Acrogenic Technologies Inc., Rockville, MD 20847, USA; dazhiyang1211@gmail.com; 4Department of Neurological Surgery, School of Medicine, Johns Hopkins University, Baltimore, MD 21205, USA

**Keywords:** RNA-sequencing, dorsal root ganglion, human birth tissue, pain, bioinformatics

## Abstract

**Background:** The biologics derived from human amniotic membranes (AMs) demonstrate potential pain-inhibitory effects in clinical settings. However, the molecular basis underlying this therapeutic effect remains elusive. HC-HA/PTX3 is a unique water-soluble regenerative matrix that is purified from human AMs. We examined whether HC-HA/PTX3 can modulate the gene networks and transcriptional signatures in the dorsal root ganglia (DRG) neurons transmitting peripheral sensory inputs to the spinal cord. **Methods:** We conducted bulk RNA-sequencing (RNA-seq) of mouse DRG neurons after treating them with HC-HA/PTX3 (15 µg/mL) for 10 min and 24 h in culture. Differential gene expression analysis was performed using the limma package, and Gene Ontology (GO) and protein–protein interaction (PPI) analyses were conducted to identify the networks of pain-related genes. Western blotting and in vitro calcium imaging were used to examine the protein levels and signaling of pro-opiomelanocortin (POMC) in DRG neurons. **Results:** Compared to the vehicle-treated group, 24 h treatment with HC-HA/PTX3 induced 2047 differentially expressed genes (DEGs), which were centered on the ATPase activity, receptor–ligand activity, and extracellular matrix pathways. Importantly, PPI analysis revealed that over 50 of these DEGs are closely related to pain and analgesia. Notably, HC-HA/PTX3 increased the expression and signaling pathway of POMC, which may affect opioid analgesia. **Conclusions:** HC-HA/PTX3 induced profound changes in the gene expression in DRG neurons, centered around various neurochemical mechanisms associated with pain modulation. Our findings suggest that HC-HA/PTX3 may be an important biological active component in human AMs that partly underlies its pain inhibitory effect, presenting a new strategy for pain treatment.

## 1. Background

Human amniotic membrane (AM), an inner part of the placenta, possesses natural anti-inflammation, anti-scarring, and anti-angiogenesis effects. One intriguing component derived from human AM is the heavy-chain hyaluronic acid/pentraxin 3 (HC-HA/PTX3), a unique water-soluble regenerative matrix [1,2]. It comprises heavy chain 1 of the inter-α-trypsin inhibitor, covalently linked to hyaluronic acid and complexed with pentraxin 3. HC-HA/PTX3 was suggested to be a key biologically active component in human AM that orchestrates multiple cellular actions related to its anti-inflammatory and anti-scarring properties [2,3]. It may also help progenitor/stem cells preserve their stem cell potency [2]. Moreover, some anecdotal clinical evidence suggests that birth tissue products containing HC-HA/PTX3 may induce pain relief in various conditions, including eye indications [4,5], osteoarthritis [6,7], and lower extremity neuropathy [8]. While HC-HA/PTX3’s anti-inflammatory and anti-scarring actions have been well documented [2,3], its neuronal mechanisms underlying pain modulation remain unexplored.

Peripheral somatic afferent inputs are transmitted by primary sensory neurons within the dorsal root ganglion (DRG). These neurons exhibit diverse neurochemical, physiological, and functional properties [9,10],and undergo dynamic changes after injury associated with persistent pain. In our recent study, an intra-paw injection of the birth tissue product FLO (Clarix Flo) and HC-HA/PTX3 attenuated behavioral pain hypersensitivity in a mouse model of post-surgical pain induced by plantar incision [11]. Notably, this pain relief aligned with a selective reduction in the activated nociceptive neurons observed in in vivo calcium imaging of the DRG [11]. Additionally, treating cultured DRG neurons with HC-HA/PTX3 acutely attenuated neuronal excitability, as evidenced by both in vitro calcium imaging and patch-clamp electrophysiology recordings, and induced cytoskeleton rearrangement. Collectively, these findings suggest that HC-HA/PTX3 can profoundly alter DRG neuron activity and functions. However, whether HC-HA/PTX3 may induce transcriptional changes in DRG neurons remains unclear.

To explore the molecular mechanisms underlying HC-HA/PTX3’s neuronal modulation effect, we used an unbiased RNA-seq method to characterize the gene response in DRG neurons following short-term (10 min) and long-term (24 h) treatment with HC-HA/PTX3 without predefined functional hypotheses. RNA-seq technology has been widely used to gain insights into the dynamics of nociceptor-specific gene expression at a genome-wide scale under different pain conditions in animal models [12]. These studies, whether through examining an entire population in bulk [12,13] or studying cell-type-specific heterogeneity using single-cell analysis [14,15,16,17], have provided important details on the molecular mechanisms that modulate pain.

We isolated a highly enriched neuronal cell population in our DRG culture system, minimizing the effects of tissue heterogeneity which is often encountered in bulk or single-cell analysis of entire DRG tissue [12,13,14,15,16]. Importantly, we used a pain interactome approach to explore the protein–protein interaction (PPI) networks in all differentially expressed genes (DEGs) after drug treatment and identify those associated with pain [12,18]. Finally, we used the transcriptome signatures to identify potential genes and identified pro-opiomelanocortin (POMC), an archetypal polypeptide precursor of hormones and neuropeptides, as an important target that may contribute to HC-HA/PTX3-induced neuronal and pain inhibition.

## 2. Methods

### 2.1. Animals

RNA-seq was conducted in wildtype C57BL/6 mice (20–30 g, Jackson Laboratory, Bar Harbor, ME, USA). In vitro calcium imaging was conducted in both wildtype C57BL/6 mice and *Pirt-MOR* cKO mice. The mice were used at 2–3 months of age and from both sexes. *Pirt-MOR* cKO mice were generated by crossing *Pirt-Cre* mice with *Oprm1^fl/fl^* mice to delete mu-opioid receptors (MORs) specifically from primary sensory neurons, as described in our previous studies [19]. In *Pirt-Cre* mice, the *Cre* recombinase is under the control of the *Pirt* promoter and expressed exclusively in ~90% of all DRG neurons. Animals were housed under optimal laboratory conditions in groups of 3–5 on a standard 12 h light/dark cycle with free access to food and water. Animal studies were approved by the Johns Hopkins University Animal Care and Use Committee (Baltimore, MD, USA) as consistent with the National Institutes of Health Guide for the Care and Use of Laboratory Animals to ensure minimal animal use and discomfort.

### 2.2. Culturing DRG Neurons for RNA-Seq

DRGs from all lumbar spinal levels of adult male and female C57BL/6 mice were collected in cold DH10 [20] [90% DMEM/F-12, 10% fetal bovine serum, penicillin (100 U/mL), and streptomycin (100 μg/mL) (Invitrogen, Waltham, MA, USA)] and treated with enzyme solution [dispase (5 mg/mL) and collagenase type I (1 mg/mL) in Hanks’ balanced salt solution without Ca^2+^ or Mg^2+^ (Invitrogen)] for 35 min at 37 °C. After trituration, the supernatant with cells was filtered through a Falcon 70 µm cell strainer. Then, the cells were spun down with centrifugation and were resuspended in DH10 with growth factors (25 ng/mL NGF; 50 ng/mL GDNF), plated on glass coverslips coated with poly-d-lysine (0.5 mg/mL; Biomedical Technologies, Stoughton, MA, USA) and laminin (10 μg/mL; Invitrogen), and cultured in an incubator (95% O_2_ and 5% CO_2_) at 37 °C. After being plated on the dish for 24 h, DRG neurons were treated with vehicle or HC-HA/PTX3 (15 µg/mL) for another 24 h.

### 2.3. RNA Isolation, Library Preparation, and Sequencing

Total RNA was isolated using a PicoPure RNA Isolation Kit (Thermo Fisher Scientific, Waltham, MA, USA) following the manufacturer’s manual. Total RNA quantity and quality were determined with a ThermoFisher Nanodrop 8000 and an Agilent Fragment Analyzer, respectively. Then, 100 ng of total RNA was input into an Illumina Total RNA prep ligation with a Ribo-Zero plus library preparation kit. Libraries were prepared following the manufacturer’s instructions. Briefly, ribosomal RNA was depleted through probe binding, followed by cleanup with RNAclean XP beads. Ribosomal-depleted RNA was fragmented and primed prior to double-stranded cDNA synthesis. Following AMPure bead cleanup, 3’ ends were adenylated, and index anchors were ligated. Libraries were amplified with UDI adapter primers. A final AMPure cleanup was performed, and libraries were verified with Qubit (Invitrogen) and a fragment analyzer for sizing and yield assessment. Libraries were pooled at equimolar ratios and sequenced on a NovaSeq 6000 SP 200 flow cell with paired-end, 2 × 100 bp reads.

### 2.4. RNA-Seq Data Analysis

Sequencing reads were aligned to the annotated RefSeq genes in the mouse reference genome (mm10) using HISAT2. Aligned reads were filtered to remove ribosomal RNA and visualized using Integrative Genomics Viewer. A gene count matrix that contained the raw transcript counts for each annotated gene was generated using the featureCounts function of the Subread package in R against the Ensemble mm10 transcriptome [21]. We filtered this matrix by removing genes with zero counts across all samples and relied on the automatic and independent filtering used by limma [22] and EdgeR [23,24] to determine the most appropriate threshold for removing genes with low counts.

To identify genes that were differentially regulated following HC-HA/PTX3 treatment, transcript counts were normalized and log2-transformed using the default normalization procedures in limma [22,25]. The technical variations between each sets/batches were minimized with the ComBat-seq package [26] or the EdgeR package [23] using the design formula (~batch + condition). The differential expression analysis was performed separately for each treatment using the default parameters. This analysis identified differentially expressed genes between the vehicle- and HC-HA/PTX3-treated groups. All downstream analyses of RNA-seq data were performed on the data obtained from limma. An adjusted *p*-value (i.e., false discovery rate [FDR]) < 0.05 and an absolute log_2_ fold change > 0.5 were used to define differentially expressed transcripts between vehicle- and HC-HA/PTX3-treated samples. Genes with differential expression between groups were then included in pathway analysis to infer their functional roles and relationships.

### 2.5. Enrichment and Protein–Protein Interaction Analysis

The analysis of the significantly enriched GO (Gene Ontology) terms over-represented among the DEG lists was performed with the ClusterProfiler package in R [27,28]. The significance of the overlap between DEG sets from the top GO terms was calculated using a hypergeometric distribution. This analysis demonstrated significant differences between most of the top DEG sets. The functional annotation clustering was performed with the TopGO package in R using the conservative Elim algorithm, which minimizes false-positive results by removing annotated genes from ancestor GO terms [29]. A PPI network was drawn using R packages igraph and ggraph based on the list of the reported 1002 PPIs involved in pain [18]. Single edges not connected to the main network were removed.

### 2.6. Preparation of HC-HA/PTX3

HC-HA/PTX3 was purified from human AM after donor eligibility was determined according to the requirements by the FDA based on our published method [30] with modifications and was performed according to good laboratory practices (GLPs). Briefly, AM was pulverized under liquid nitrogen (Freezer/Mill, Spex^®^ SamplePrep, Metuchen, NJ, USA), extracted with PBS (pH 7.4) at 1:1 (mL/g) at 4 °C for 1 h, and centrifuged at 48,000× *g* at 4 °C for 30 min to generate AM extract (AME). The solid cesium chloride (CsCl) and 8 M guanidine-HCl/PBS (GnHCl) with protease inhibitors (at a final concentration of 10 mM aminocaproic acid, 10 mM EDTA, 10 mM N-ethylmaleimide, 1 mM phenylmethylsulfonyl fluoride (PMSF)) were added to AME and mixed at room temperature to achieve 4 M GnHCl and a density of 1.35 g/mL before centrifugation at 125,000× *g* at 15 °C for 66 h (SW 32 Ti Swinging-Bucket Rotor, Beckman Coulter, Inc., Brea, CA, USA). After subdivided into 12 fractions (3 mL/fraction) from top to bottom per tube, the hyaluronic acid (HA) in the samples from each fraction was determined using a HA quantitation assay (Corgenix, Inc., Broomfield, CO, USA) and the proteins using a BCA protein assay (Thermo Fisher Scientific, Waltham, MA USA). After excluding fractions 1 and 2, which contained proteins but no HA, the remaining fractions were pooled, adjusted with CsCl, 8 M GnHCl, and PBS to 4 M GnHCl and a density of 1.40 g/mL, then underwent the second ultracentrifugation under the same conditions. The third and fourth ultracentrifugations were processed similarly except the starting density was 1.42 g/mL. After each ultracentrifugation, fractions 1 and 2 were discarded, and, at the last ultracentrifugation, fractions 3 to 9 were pooled, dialyzed (dialysis tubing, 3 kD MWCO) extensively against filtered water, lyophilized (SPScientific AdVantage Pro XL Freeze Dryer/Lyophilizer, SP Scientific, Warminster, PA, USA), and stored at −80 °C before use. A validation study confirmed a shelf life of 4 years.

### 2.7. Immunoblotting

The cultured neurons from each dish were lysed in ice-cold radioimmunoprecipitation assay (RIPA) buffer (Sigma, St. Louis, MO, USA) containing a protease/phosphatase inhibitor cocktail (Cell Signaling Technology, Boston, MA, USA). The protein concentration of RIPA lysates was determined by a standard bicinchoninic acid protein assay (Thermo Fisher Scientific, Waltham, MA, USA). Samples (20 µg) were separated on a 4% to 12% Bis-Tris Plus gel (Thermo Fisher Scientific) and then transferred onto a polyvinylidene difluoride membrane (Thermo Fisher Scientific). Immunoreactivity was detected by enhanced chemiluminescence (ECL; Bio-Rad, Hercules, CA, USA) after incubating the membranes with the indicated primary antibody (4 °C, overnight). Antibodies were chosen based on previous findings and our own study. GAPDH (EMD Millipore, 1:100,000) was used as an internal control for protein loading. We used primary antibodies against TRPA1 (Novus Biologicals, NB110-40763SS; 1:2000), TRPV1 (Novus Biologicals, NB100-1617; 1:2000), and POMC (Cell Signaling Technology, 23499S; 1:1000) were used as positive controls. Western blots were imaged with an ImageQuant LAS 4000 (GE Healthcare Life Sciences), and ImageJ (ImageJ 1.46r, LOCI, University of Wisconsin, Madison, WI, USA) was used to quantify the intensity of the immunoreactive bands of interest from autoradiograms.

### 2.8. In Vitro Calcium Imaging

Experiments were conducted as described in our previous studies [17,31]. Neurons were loaded with the fluorescent calcium indicator Fura-2-acetomethoxyl ester (2 μg/mL, Molecular Probes, Eugene, OR, USA) for 45 min in the dark at room temperature and then allowed to de-esterify for 15 min at 37 °C in a warm external solution. After being washed, cells were imaged at 340 and 380 nm excitation for the detection of intracellular free calcium. At the end of the experiment, KCl (50 mM) was added to confirm cell viability.

### 2.9. Statistical Analysis

Statistical analyses were performed with the Prism 9.0 statistical program (GraphPad Software, Boston, MA, USA). The methods for statistical comparisons in each study are indicated in the figure legends. To reduce selection and observation bias, animals were randomized to the different groups, and the experimenters were blinded to drug treatment. The comparisons of data consisting of two groups were made with Student’s *t*-test. Comparisons of data in three or more groups were made with one-way analysis of variance (ANOVA), followed by the Bonferroni post hoc test. Two-tailed tests were performed, and *p* < 0.05 was considered statistically significant in all tests.

## 3. Results

### 3.1. HC-HA/PTX3 Treatment of Cultured DRG Neurons for RNA-Sequencing

DRG neurons were harvested from naïve mice and plated on a dish for 24 h [22,24,25,32], followed by treatment with vehicle or HC-HA/PTX3 (15 µg/mL), based on our preliminary findings, for 10 min or for another 24 h. The bulk RNA-seq transcriptome profiles were then measured in both groups (Figure 1A). As an additional control for the specificity of our neuronal culture approach, we analyzed the relative expression of a subset of marker genes that are known to be enriched specifically in different cell types in the DRG [16,33]. By comparing our RNA-seq with single-cell RNA-seq datasets from existing sources [16], we found enrichment of neuron-specific markers, such as *Calca* (CGRPα), *Scn10a* (Nav1.8), or *Tubb3,* in the mRNAs isolated from our neuronal culture. In contrast, glial marker genes (*Apoe*, *Fabp7*) and endothelial-specific genes (*Cldn5*, *Flt1*) showed very low levels of expression (Figure 1B). Thus, we identified a highly enriched neuronal population that minimized the effects of cell heterogeneity in bulk RNA-seq using whole DRG tissue.

To identify gene expression changes, we performed unbiased RNA-seq and measured all poly(A)-containing transcripts in the neurons. Because the plotting of the gene loadings showed some batch effects, we then minimized the technical variations between each batch during the differential expression analysis by using the ComBat-seq package or the EdgeR package, which have been commonly used in previous studies [22,23,26]. The PCA plot showed a clear separation between the vehicle- and 24 h HC-HA/PTX3-treated groups (Figure 1C).

### 3.2. HC-HA/PTX3 Treatment (24 h) Broadly Changed the Gene Expressions in DRG Neurons

The expression levels of 16,979 genes were evaluated, and no significant differences in the expression levels between the vehicle and acute HC-HA/PTX3 treatment (10 min) were detected. In contrast, 2047 genes showed significant differences in expression levels between the 24 h HC-HA/PTX3 and vehicle groups. Of these, 1038 (50.7%) genes were downregulated and 1009 (49.3%) were upregulated by HC-HA/PTX3 (Figure 2A). Lists of these genes with increased and decreased expressions are provided in Appendix A. The Venn diagram showed a substantial overlap between the DEGs following batch effect correction by using either ComBat-seq (red) or edgeR (blue, Figure 2B), suggesting that this approach was consistent and reliable in adjusting the batch effect. The log_2_FC of the 200 most variable genes in the 24 h HC-HA/PTX3 group, as compared to the vehicle group, is shown in a heat map (Figure 2C).

To gain insight into the potential functional properties of DEGs induced by 24 h HC-HA/PTX3 treatment (genes from Appendix A identified after ComBat-seq adjustment), we performed an enrichment analysis of gene ontology (GO) terms. The analysis of the functional clusters using ClusterProfiler [27,28] showed that ATPase activity, receptor-ligand activity, and extracellular matrix were among the highly represented GO terms in “molecular function” (Figure 2D). The analysis also showed over-representations of extracellular matrix-related and cytoskeleton-related GO terms in the top category of “cellular component” (Figure 2E). Gene clustering in the top 10 GO terms was further visualized as a circular dendrogram (Appendix A). These findings of DRG transcriptional signatures are in agreement with the functions of HC-HA/PTX3 observed in other cell types [3]. For example, HC-HA/PTX3 downregulates the expression of bone morphogenic protein 4 (*Bmp4*) and *Bmp7,* and upregulates the expression of *Bmp8a* in DRG neurons in our dataset. Our previous studies showed that HC-HA/PTX3 upregulates BMP signaling in limbal niche cells (LNCs), which supports the quiescence and self-renewal of limbal basal epithelial progenitor cells (LEPCs) [34]. Moreover, HC-HA/PTX3 activates BMP signaling to revert human corneal fibroblasts and myofibroblasts to keratocytes [3].

### 3.3. HC-HA/PTX3 Treatment Altered the Expression of Pain-Related Genes

In order to explore the DEGs that may be associated with pain, we conducted protein–protein interaction (PPI) analysis of the 2047 DEGs that were significantly regulated by HC-HA/PTX3 to identify the networks of pain-related genes. We used the pain interactome, a comprehensive network of 611 interconnected proteins associated with pain [18], and identified a core pain-related PPI network that consists of 55 genes (23 upregulated and 32 downregulated, Figure 3A) linked to 140 interaction partners (genes that are not from the list of 2047 DEGs).

The key hub gene of this network was upregulated POMC [35]. In addition to an upregulated POMC pathway, the network contained other pain-related changes, which could be divided into five major categories [36,37], including peptides (e.g., *Ghrh*, *Nppb*, *Ntf3*), cytokines (e.g., *Bmp7*, *Ccl2*, *Ccr5*), G-protein coupled receptors (GPCRs, e.g., *Crhr1*, *Gnrhr*, *Sstr1*), ECM components (e.g., *Lama1*, *Mmp2*, *Mmp9*), and others (Figure 3B). These DEGs center on the mediators of the neurochemical mechanisms of pain. The pain-inhibitory changes included upregulated anti-nociceptive genes, such as *Nppb* [38], *Sst* [39], *Gdf15* [40], *Il7* [41], and *Il11* [42], and downregulated pro-nociceptive genes, including *Ccl2* [43], *Ccr5* [44], *Cxcl5* [45], *Mmp9* [46], *Mmp13* [46,47] and *Ephb1* [48]. However, some pro-nociceptive genes were also upregulated, including *Tac1* [49], *Tacr1* [50], *Mmp2* [46,51], and *Casp1* [52], and anti-nociceptive gene Sstr1 was downregulated [53]. The complex changes in gene expressions at the protein level and their net effect on pain warrant further investigation.

### 3.4. The Functional Upregulation of POMC May Contribute to Neuronal Inhibition

We further examined the interconnected network of the 53 genes interacting with POMC. The hub genes *Sst*, *Tmpo*, *Crhr1*, and *Bmp7* showed significant changes (Figure 3C). POMC produces the opioid peptide β-endorphin, an essential endogenous agonist that activates its cognate MOR for pain inhibition [35,54]. Importantly, 24 h HC-HA/PTX3 treatment upregulated the crucial hub gene POMC, accompanied by increased protein expression (Figure 3D), whereas TRPV1 and TRPA1 expressions were unchanged (Appendix A).

To examine the functional implications of upregulated POMC expression, we conducted in vitro calcium imaging of DRG neurons following 24 h of HC-HA/PTX3 treatment. Our findings showed a significant decrease in the percentage of capsaicin-responsive neurons and peak calcium responses in DRG neurons after HC-HA/PTX3 treatment compared to the control group (Figure 4A,B). Furthermore, this effect of long-term HC-HA/PTX3 treatment was blocked by co-treatment with CTOP (a highly selective MOR antagonist, Figure 4A,B) and was absent from the DRG neurons of *Pirt-MOR* cKO mice (Figure 4C,D).

## 4. Discussion

Extracts and particulates of the human AM and umbilical cord have been applied in a wide range of conjunctival and corneal conditions [34,55]. HC-HA/PTX3, the purified matrix component derived from human AM, has recently gained our attention as a potential solution for pain. Here, we found 2047 DEGs and changes in distinct gene networks in DRG neurons after 24 h of treatment with HC-HA/PTX3 by using unbiased bulk RNA-seq. Intriguingly, over 50 of these DEGs are closely related to pain and analgesia, as evidenced by PPI analysis. Among them, we further confirmed the expression and functional upregulation of POMC signaling, involved in opioid analgesia, as a proof of concept. These findings suggest that long-term treatment with HC-HA/PTX3 induced profound changes in the gene expression within DRG neurons. These changes centered around various neurochemical mechanisms associated with pain modulation, which may partly underlie pain inhibition by human AM and present a new strategy for pain treatment.

By comparing our RNA-seq dataset with single-cell RNA-seq data from existing sources [10,14,56], we found that neuron-specific markers, but not glial marker genes or endothelial-specific genes, were enriched in the mRNAs isolated from our neuronal culture. Thus, the top-ranked DEGs likely reflect those in DRG neurons that mediate the sensory inputs from the periphery to the spinal cord. They detect various sensory stimuli (e.g., heat, cold, tactile, chemicals), which are transduced by different receptors and ion channels on their plasma membranes. Small DRG neurons are activated by heat and noxious stimulation, while innocuous mechanical stimulation mainly activates large DRG neurons and low-threshold mechanoreceptors (LTMRs) [57]. Functionally, 24 h HC-HA/PTX3 treatment decreased the percentage of capsaicin-responsive DRG neurons and the peak calcium responses, pointing to long-lasting changes in neuronal function that often involve transcriptional and translational mechanisms. It remains to be determined whether HC-HA/PTX3 induces differential changes in functional distinct subtypes of DRG neurons. Moreover, sex dimorphism in pain processing has been well documented. Although our separate study found that HC-HA/PTX3 induced comparable behavioral pain inhibition in both sexes [11], it remains possible that HC-HA/PTX3 could cause differential transcriptional and functional changes in DRG neurons from male and female animals. Additionally, there may be complex interplay between estrogen and HC-HA/PTX3 signaling in sensory neurons, warranting further investigation.

Notably, 24 h HC-HA/PTX3 treatment induced profound changes in the expression of genes centered on the neuroactive ligand–receptor interaction pathway and neurochemical mechanisms of pain in the DRG. In particular, POMC upregulation is among the hub genes changed by HC-HA/PTX3 that are important to neuronal and pain inhibition. As a proof of principle, we validated increased POMC expression at the protein level and its functional impact using in vitro calcium imaging. POMC is responsible for producing the opioid peptide β-endorphin, which acts as an endogenous agonist of opioid receptors, with a particular affinity for MOR [35,54]. Indeed, our findings that CTOP, a highly selective MOR antagonist, blocked the decreased percentage of capsaicin-responsive neurons induced by 24 h HC-HA/PTX3 treatment suggested a functional upregulation of POMC-induced MOR signaling in DRG neurons, which may achieve pain inhibition. Further investigations into the changes in β-endorphin expression in DRG neurons and the concentration of the released β-endorphin in the culture medium following HC-HA/PTX3 treatment are warranted. Additionally, future behavioral assays in appropriate animal pain models should be conducted to provide direct in vivo evidence of the role of the POMC signaling in the pain-inhibitory effects of HC-HA/PTX3.

Unlike common analgesics (e.g., lidocaine, gabapentin), which only target one down-stream effector (e.g., sodium channels, calcium channels) and are short-lasting, HC-HA/PTX3-induced DEGs are enriched in regulating multiple cellular functions. These changes involve genes that regulate receptor ligand activity, the ECM, and the cytoskeleton. Thus, we postulate that this naturally occurring biologic may induce broad changes and render a prolonged decrease in nociceptive neuronal excitability. In this regard, comparing the transcriptional changes induced by HC-HA/PTX3 with those caused by traditional analgesics such as morphine and gabapentin in DRG neurons would be meaningful. This comparison could help better understand and differentiate their pain-inhibition mechanisms.

In the current study, we treated DRG neurons with HC-HA/PTX3 for 10 min and 24 h. Despite the minimal gene expression changes between short-term HC-HA/PTX3 treatments (10 min) and the vehicle, HC-HA/PTX3 induced acute neuronal and pain inhibitory effects, including the inhibition of cell membrane ion channels via a CD44-mediated cytoskeletal rearrangement in DRG neurons, as demonstrated in our recent study [11]. Therefore, HC-HA/PTX3 could induce both acute and prolonged functional changes in DRG neurons, affecting neuronal excitability and pain transmission, depending on the duration of treatment. The receptor mechanisms and intracellular signaling cascades that mediate the profound transcriptional changes after prolonged HC-HA/PTX3 treatment remain to be determined, such as by using CD44 knockout mice.

Repeated and long-term drug treatments, such as with morphine, are often necessary for managing chronic pain. However, tachyphylaxis and analgesic tolerance frequently develop after repeated drug use, limiting their clinical effectiveness. Our findings suggest that HC-HA/PTX3 profoundly alters the transcription profile of DRG neurons, involving genes that regulate receptor ligand activity, the ECM and cytoskeleton, and POMC signaling, potentially inducing long-term changes in neuronal excitability. Thus, it is important to test if repeated HC-HA/PTX3 treatment in animal pain models leads to prolonged pain inhibition without tolerance in the future. Moreover, we previously demonstrated that HC-HA/PTX3 exerts anti-inflammatory and anti-scarring effects that promote regenerative healing [2] and helps maintain stem cell quiescence [34], supporting tissue homeostasis and repair. Since non-resolving inflammation and delayed wound healing sensitize neurons and contribute to pain chronification, HC-HA/PTX3 may help limit local inflammatory responses, thereby facilitating wound healing and promoting pain resolution. These actions may serve as indirect mechanisms of pain inhibition, warranting further investigation.

## 5. Conclusions

In summary, our study found distinct gene networks and transcriptional signatures in mouse DRG neurons that are altered by HC-HA/PTX3 from human AM. These findings suggest that HC-HA/PTX3 holds promise not only for regenerative medicine but also as a player in pain modulation. Further investigations into its precise mechanisms could identify novel therapeutic avenues for HC-HA/PTX3 to be deployed as a viable biologic for pain treatment.

## Figures and Tables

**Figure 1 cells-13-01887-f001:**
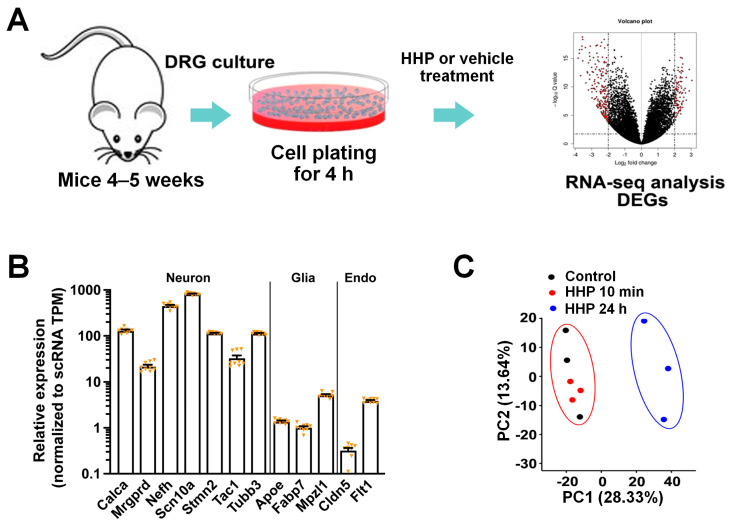
**HC-HA/PTX3 (HHP) treatment of dorsal root ganglia (DRG) neurons for RNA-sequencing.** (**A**) Schematic diagram of experimental procedure. RNA-seq was performed on cultured wildtype (WT) mouse DRG neurons treated with vehicle or HC-HA/PTX3 (15 µg/mL) for 10 min or 24 h. (**B**) Neuronal markers, such as *Calca* (CGRP) or *Tubb3*, had high-level expression in the RNA samples from cultured DRG neurons, whereas those for glial (*Apoe*, *Fabb7*) and endothelial (*Cldn5*, *Fit1*) markers were much lower. N = 9. Data are mean ± SEM. (**C**) Principal component analysis (PCA) of the samples treated with vehicle or HC-HA/PTX3 (10 min, 24 h). PCAs were generated after batch effect correction by applying ComBat-seq.

**Figure 2 cells-13-01887-f002:**
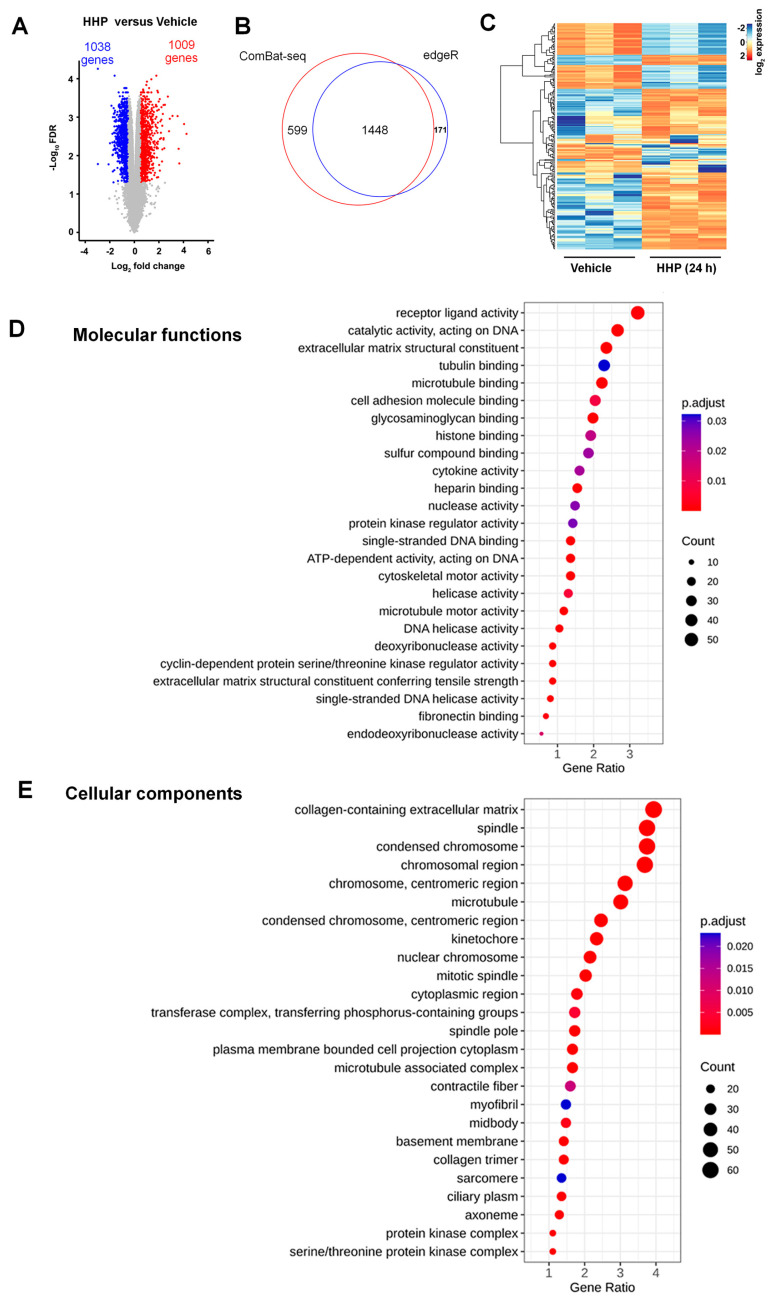
**HC-HA/PTX3 (HHP) treatment for 24 h broadly changed gene expression in dorsal root ganglia (DRG) neurons, which may affect molecular functions and cellular components.** (**A**) Volcano plot of differentially expressed genes (DEGs) in cultured wildtype (WT) DRG neurons after vehicle or 24 h of HC-HA/PTX3 treatment (15 µg/mL). DEGs were identified by a [log2 fold change (FC)] > 0.5 and an FDR < 0.05. Significant downregulated and upregulated genes are designated in blue and red colors. (**B**) Venn diagram representing the number of DEGs identified after batch effect correction by ComBat-seq (red), edgeR (blue), and in both (overlap). (**C**) Heatmap shows the log2FC of the 200 most variable genes in 24 h HC-HA/PTX3-group as compared to the vehicle group. (**D**,**E**) Gene Ontology (GO) enrichment analysis of DEGs induced by HC-HA/PTX3. ClusterProfiler analysis of significantly enriched GO terms within molecular function categories (**D**) and cellular component categories (**E**). FDR-adjusted *p* < 0.05.

**Figure 3 cells-13-01887-f003:**
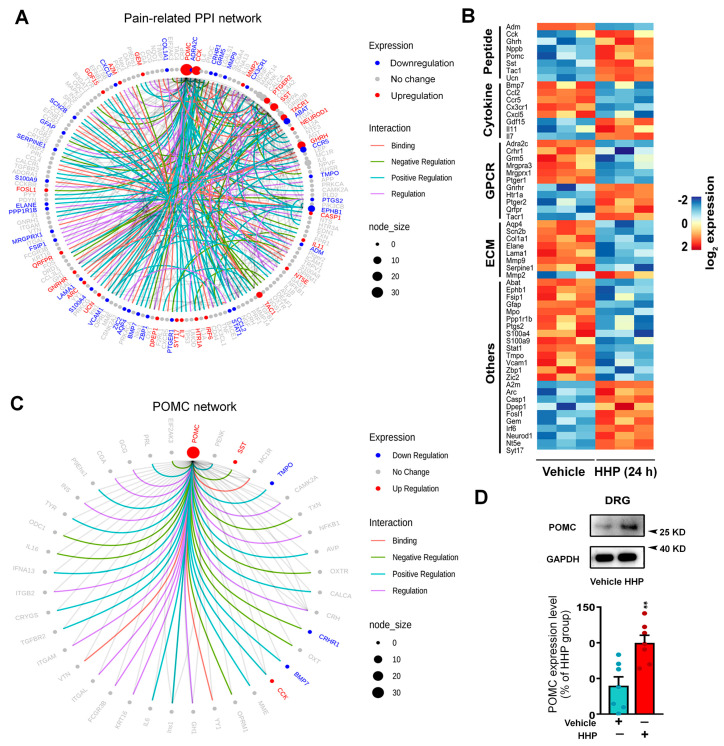
**Long-term HC-HA/PTX3 treatment altered the expression of pain-related genes in dorsal root ganglia (DRG) neurons.** (**A**) The pain-related PPI network of the DEGs in cultured wildtype (WT) mouse DRG neurons after 24 h of HC-HA/PTX3 treatment (15 µg/mL) as compared to the vehicle. The network was built based on the pain interactome using DEGs, together with neighbors from the pain interactome network (DEGs and non-DEGs). (**B**) Heatmap shows the fold change of DEGs in each biological replicate of vehicle- and HC-HA/PTX3-treated samples (15 µg/mL, 24 h). The genes shown on the heatmap that encode neuropeptides, cytokines, GPCRs, extracellular matrix (ECMs) were significantly regulated by HC-HA/PTX3. (**C**) The pro-opiomelanocortin (POMC) network of DEGs induced by 24 h treatment with HC-HA/PTX3 (15 µg/mL) in WT DRG neurons. Colored edges mark the type of interaction. Colored nodes mark the expression changes (up/down/no change) after HC-HA/PTX3 treatment. Node size indicates the number of interactions against pain interactome, as explained in the legend. (**D**) Western immunoblotting (cropped blots, full-length blots are presented in Appendix A) shows an upregulation of POMC expression after 24 h of HC-HA/PTX3 treatment (15 µg/mL). The quantification of POMC protein levels (28 kDa), which were normalized to GAPDH (37 kDa). The mean POMC level in the HC-HA/PTX3 group was considered to be 100%. N = 6 mice/group. Data are mean ± SEM. Unpaired Student’s *t*-test. t = 3.69, df = 12, ** *p* < 0.01.

**Figure 4 cells-13-01887-f004:**
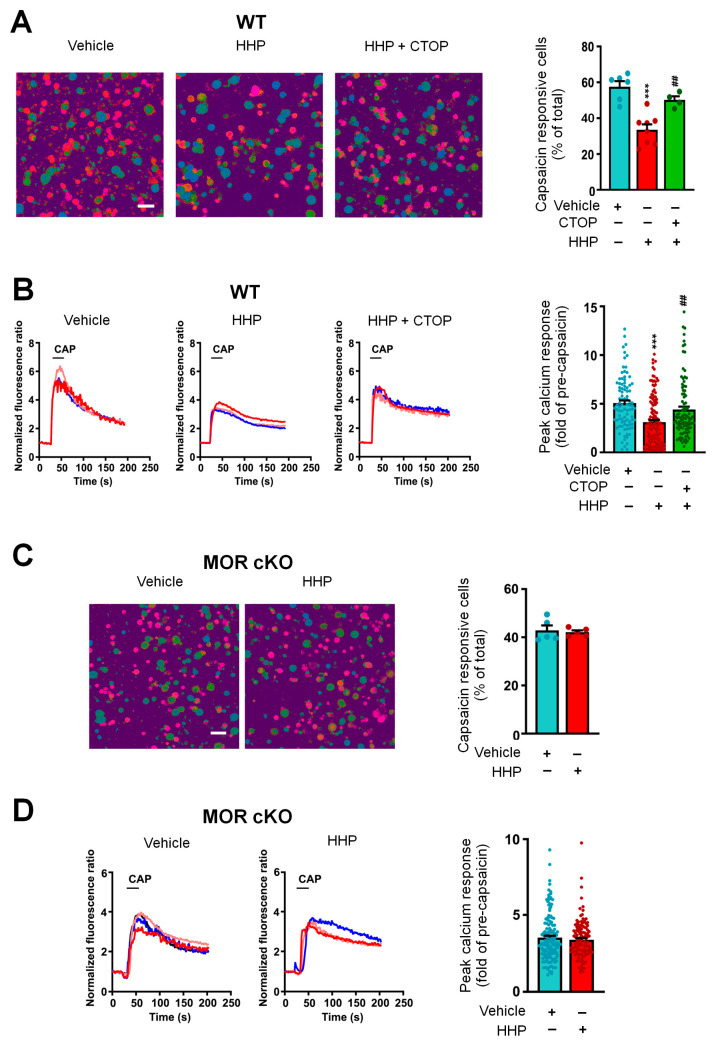
**Long-term HC-HA/PTX3 treatment attenuated capsaicin-evoked responses in dorsal root ganglia (DRG) neurons, which involved increased endogenous pro-opiomelanocortin (POMC) signaling.** (**A**) Left: Images showing calcium response of cultured wildtype (WT) mouse DRG neurons to capsaicin (CAP, 0.3 μM, 20 s), which are marked in red. Scale bars, 50 μm. Right: The quantification shows that the percentage of capsaicin-responsive neurons was significantly decreased by HC-HA/PTX3 (15 µg/mL, 24 h). This effect of HC-HA/PTX3 was blocked by co-treatment with CTOP [a mu-opioid receptor (MOR) antagonist,10 nM, 24 h] with HC-HA/PTX3. N = 4–8 mice/group. Data are mean ± SEM. One-way ANOVA followed by Bonferroni post hoc test. F_(2,15)_ = 19.22, *** *p* < 0.001 versus vehicle. ^##^
*p* < 0.01 versus HC-HA/PTX3. (**B**) Left: Representative traces show that capsaicin (0.3 μM, 20 s) evoked an increase in [Ca^2+^]i in WT mouse DRG neurons, which was significantly reduced by HC-HA/PTX3 (15 µg/mL, 24 h) but not in those co-treated with CTOP. Right: Quantification of responses of individual neurons in each group, as shown in A. Data are mean ± SEM. One-way ANOVA followed by Bonferroni post hoc test. F_(2, 326)_ = 15.18, *** *p* < 0.001 versus vehicle. ^##^
*p* < 0.01 versus HC-HA/PTX3. (**C**) Left: Images showing calcium response of DRG neurons from MOR conditioning knockout (cKO) mice to capsaicin (0.3 μM, 20 s). Scale bars, 50 μm. Right: The percentage of capsaicin-responsive neurons following HC-HA/PTX3 (15 µg/mL, 24 h) or vehicle treatment. N = 5 mice/group. Data are mean ± SEM. Unpaired Student’s *t*-test. t = 0.34, df = 8, *p* > 0.05. (**D**) Representative traces show that capsaicin (0.3 μM, 20 s) evoked an increase in [Ca^2+^]i in DRG neurons from MOR cKO mice. Right: Quantification of calcium responses to capsaicin of individual MOR cKO DRG neurons from the vehicle-treated and HC-HA/PTX3-treated groups, as shown in C. Data are mean ± SEM. Unpaired Student’s *t*-test. t = 0.82, df = 260, *p* > 0.05.

## Data Availability

RNA-sequencing data were deposited in GEO under accession number GSE242389, accessed on 25 June 2024 (https://www.ncbi.nlm.nih.gov/geo/query/acc.cgi?acc=GSE242389).

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
