# Peer review of "HC-HA/PTX3 from Human Amniotic Membrane Induced Differential Gene Expressions in DRG Neurons: Insights into the Modulation of Pain"

_cells, 2024, doi:10.3390/cells13221887_

Round 1
Reviewer 1 Report
Comments and Suggestions for Authors
In this paper, He and colleagues examined the effect of HC-HA/PTX3 (a water-soluble regenerative matrix) on genomic effects in mouse DRG neurons. The neurons were incubated acutely (10 min) or chronically (24 hr). The results indicate that roughly 2047 genes were either up- or down-regulated following the chronic exposure to the matrix for 24 hr, while the 10 min incubation time was without significant effects. The authors applied protein-protein interaction (PPI) analysis to examine the relationship of the 2047 genes to pain pathways, and their results showed there were approximately 611 interconnected proteins of which 23 genes were upregulated and 32 were downregulated. Their results also showed the POMC pathway was a key hub gene. One functional assay showed that the 24 hr treatment decreased the capsaicin-evoked responses in DRG neurons. Overall, the paper was well written and logically presented as it builds on prior work by the authors. One major concern with the manuscript is that the discussion section was a bit short and appeared to primarily repeat the results of the study. It would benefit the readers to get some insight into what the results mean and how they fit with what is currently known and what needs to be explored.
Some minor concerns:
1. Line 71: uses future tense
2. Figures 2D, 2E, 3A and 3C are blurry and difficult to read.
3. Formatting of text is inconsistent: Lines 201-207
Author Response
We appreciate the editor and reviewers’ thoughtful comments and helpful suggestions on our submission! In this revision, we include more information and have carefully revised the manuscript according to reviewers’ suggestions. Major changes are marked in red. We provide below a point-by-point response to the reviewers’ comments.
Reviewer 1
Comments and Suggestions for Authors
In this paper, He and colleagues examined the effect of HC-HA/PTX3 (a water-soluble regenerative matrix) on genomic effects in mouse DRG neurons. The neurons were incubated acutely (10 min) or chronically (24 hr). The results indicate that roughly 2047 genes were either up- or down-regulated following the chronic exposure to the matrix for 24 hr, while the 10 min incubation time was without significant effects. The authors applied protein-protein interaction (PPI) analysis to examine the relationship of the 2047 genes to pain pathways, and their results showed there were approximately 611 interconnected proteins of which 23 genes were upregulated and 32 were downregulated. Their results also showed the POMC pathway was a key hub gene. One functional assay showed that the 24 hr treatment decreased the capsaicin-evoked responses in DRG neurons. Overall, the paper was well written and logically presented as it builds on prior work by the authors.
One major concern with the manuscript is that the discussion section was a bit short and appeared to primarily repeat the results of the study. It would benefit the readers to get some insight into what the results mean and how they fit with what is currently known and what needs to be explored.
Response: We thank the reviewer for these good comments and support! We agree with your suggestions and have provided more information about the implications of these findings, future studies to extend current findings, and more discussions also to address the other reviewer’s comments.
Some minor concerns:
- Line 71: uses future tense
Response: corrected.
- Figures 2D, 2E, 3A and 3C are blurry and difficult to read.
Response: We remade the figure and increased the size of these panels, which can also be seen in high resolution TIFF files submitted to the journal.
- Formatting of text is inconsistent: Lines 201-207
Response: corrected.
Reviewer 2 Report
Comments and Suggestions for Authors
ID Cells-3303022: “HC-HA/PTX3 from human amniotic membrane induced differential gene expressions in DRG neurons: Insights into the modulation of pain.”
In this study, the authors aimed to investigate the effects of HC-HA/PTX3 on gene expression in DRG neurons involved in pain signaling. They identified 2047 differentially expressed genes in mouse DRG neurons after 24 hours of HC-HA/PTX3 treatment. Moreover, they identified key pathways affected including ATPase activity and receptor-ligand interactions, with a notable enhancement of the POMC signaling pathway associated to opioid analgesia. The authors concluded that HC-HA/PTX3 induces significant changes in gene expression within DRG neurons, underscoring its potential for developing targets for pain treatment. Most relevant achievements to the topic are well-discussed.
Per my comment above – the manuscript needs minor corrections and clarifications:
(1) Abstract (page 1): The abstract is organized into sections but lacks an “methods” section. Please revise to include this information.
(2) The manuscript would benefit with the inclusion of an abbreviations section.
(3) Methods section: Ensure descriptive statistics match the data type. Was the data tested for normal distribution? Report exact ANOVA F-values, degree of freedom and post hoc test of applicable. Also, the manuscript states that two-way ANOVA was applied, but results using this method were not evident. Given the relatively low sample size, was a power analysis conducted to verify adequacy in detecting statistical differences with sufficient power?
(4) Figure 1 (Page 6, panel B): Clarify if histograms are represented as mean and SD or SEM.
(5) Supplementary Figures: Supplemental Figures 1 and 2 (Pages 8 and 10) should be moved to the appendix rather than the main manuscript.
(6) The study uses samples from both male and female animals. Given well-documented sex-based differences in pain processing, is there any anticipated impact on results due to this limitation?
(7) The discussion lacks the inclusion of limitations. What outcomes might arise from varied or prolonged HC-HA/PTX3 exposure? A comparison between HC-HA/PTX3 and traditional analgesics in pain inhibition mechanisms would be valuable.
Author Response
We appreciate the editor and reviewers’ thoughtful comments and helpful suggestions on our submission! In this revision, we include more information and have carefully revised the manuscript according to reviewers’ suggestions. Major changes are marked in red. We provide below a point-by-point response to the reviewers’ comments.
Reviewer 2
Comments and Suggestions for Authors
ID Cells-3303022: “HC-HA/PTX3 from human amniotic membrane induced differential gene expressions in DRG neurons: Insights into the modulation of pain.”
In this study, the authors aimed to investigate the effects of HC-HA/PTX3 on gene expression in DRG neurons involved in pain signaling. They identified 2047 differentially expressed genes in mouse DRG neurons after 24 hours of HC-HA/PTX3 treatment. Moreover, they identified key pathways affected including ATPase activity and receptor-ligand interactions, with a notable enhancement of the POMC signaling pathway associated to opioid analgesia. The authors concluded that HC-HA/PTX3 induces significant changes in gene expression within DRG neurons, underscoring its potential for developing targets for pain treatment. Most relevant achievements to the topic are well-discussed.
Response: We thank the reviewer for these good comments and support!
Per my comment above – the manuscript needs minor corrections and clarifications:
(1) Abstract (page 1): The abstract is organized into sections but lacks an “methods” section. Please revise to include this information.
Response: We thank the reviewer for this comment. We added “Methods” to the “Abstract” in the revised manuscript.
(2) The manuscript would benefit with the inclusion of an abbreviations section.
Response: We thank the reviewer for this suggestion. We included an “Abbreviations” section after the “Abstract” section in the revised manuscript.
(3) Methods section: Ensure descriptive statistics match the data type. Was the data tested for normal distribution? Report exact ANOVA F-values, degree of freedom and post hoc test of applicable. Also, the manuscript states that two-way ANOVA was applied, but results using this method were not evident. Given the relatively low sample size, was a power analysis conducted to verify adequacy in detecting statistical differences with sufficient power?
Response: We thank the reviewer for pointing out this mistake which was overlooked. We removed two-way ANOVA from the method, which was not applied to this study. We also added more information (method, t value, f value) about statistic analysis in the figure legend. We did not perform a power analysis. The sample size was based on previous similar studies, which were usually 3 for RNAseq and 3-5 for WB studies.
(4) Figure 1 (Page 6, panel B): Clarify if histograms are represented as mean and SD or SEM.
Response: Data in Figure 1B are represented as mean ± SEM.
(5) Supplementary Figures: Supplemental Figures 1 and 2 (Pages 8 and 10) should be moved to the appendix rather than the main manuscript.
Response: Corrected.
(6) The study uses samples from both male and female animals. Given well-documented sex-based differences in pain processing, is there any anticipated impact on results due to this limitation?
Response: We appreciate reviewer for this question, we added a brief discussion about potential sex difference and future studies to further investigate this matter.
(7) The discussion lacks the inclusion of limitations. What outcomes might arise from varied or prolonged HC-HA/PTX3 exposure? A comparison between HC-HA/PTX3 and traditional analgesics in pain inhibition mechanisms would be valuable.
Response: We appreciate reviewer for this question, and added more discussion and included limitation and remaining questions to be examined in future.